Prediction of sporulating Firmicutes from uncultured gut microbiota using SpoMAG, an ensemble learning tool

Terra Machado Douglas
Bernardes Brustolini Otávio José
dos Santos Corrêa Ellen
Ribeiro Vasconcelos Ana Tereza atrv@lncc.br
Laboratório de Bioinformática, Laboratório Nacional de Computação Científica , Quitandinha, Petrópolis , Rio de Janeiro , Brazil
Wang Liang
Electronic publication date: 2025 Oct 17
Publication date: 2025
Volume: 13
Electronic Location ID: e20232
Received 2025 Jul 11; Accepted 2025 Sep 23
Copyright: ©2025 Terra Machado et al.
Copyright year: 2025
Copyright holder: Terra Machado et al.
License: This is an open access article distributed under the terms of the Creative Commons Attribution License, which permits unrestricted use, distribution, reproduction and adaptation in any medium and for any purpose provided that it is properly attributed. For attribution, the original author(s), title, publication source (PeerJ) and either DOI or URL of the article must be cited.
License URL: https://creativecommons.org/licenses/by/4.0/

Keywords: Bacteria sporulation, Metagenome-assembled genomes, Firmicutes, Machine learning, Spore-forming bacteria

Funding: Carlos Chagas Filho Foundation for Research Support of the State of Rio de Janeiro (FAPERJ) E-26/201.046/2022 and E-26/211.107/2021 National Council for Scientific and Technological Development (CNPq) 307145/2021-2 CNPq 177392/2024-0 This study was developed in the frameworks of Carlos Chagas Filho Foundation for Research Support of the State of Rio de Janeiro (FAPERJ) (E-26/201.046/2022 and E-26/211.107/2021) and supported by the National Council for Scientific and Technological Development (CNPq) (Process number: 307145/2021-2). This study is part of the requirements for Douglas Terra Machado’s doctoral degree in Genetics at the Universidade Federal do Rio de Janeiro. Douglas Terra Machado was supported by a scholarship from CNPq (Process number: 177392/2024-0). The funders had no role in study design, data collection and analysis, decision to publish, or preparation of the manuscript.

==============================
Sporulation represents a key adaptive strategy among Firmicutes, facilitating bacterial persistence under environmental stress while mediating host colonization, transmission dynamics, and microbiome stability. Despite the recognized ecological and biomedical significance of spore-forming Bacilli and Clostridia, most taxa remain uncultivated, limiting phenotypic characterization of their sporulation capacity. To bridge this knowledge gap, we developed SpoMAG, an ensemble machine learning framework that predicts sporulation potential of metagenome-assembled genomes (MAGs) through supervised classification models trained on the presence/absence of 160 sporulation-associated genes. This R-based tool integrates Random Forest and support vector machine algorithms, achieving probabilistic predictions with high performance (AUC = 92.2%, F1-score = 88.2%). Application to fecal metagenomes from humans, cattle, poultry, and swine identified 63 putatively spore-forming MAGs exhibiting distinct host- and order-specific patterns. Bacilli MAGs from Bacillales and Paenibacillales orders showed high sporulation probabilities and gene richness, while Clostridia MAGs exhibited more heterogeneous profiles. Predictions included undercharacterized families in the spore-forming perspective, such as Acetivibrionaceae, Christensenellaceae, and UBA1381, expanding the known phylogenetic breadth of sporulation capacity. Nine genes were consistently present across all predicted spore-formers (namely pth, yaaT, spoIIAB, spoIIIAE, spoIIIAD, ctpB, ftsW, spoVD, and lgt), suggesting conserved genetic elements across uncultivated Firmicutes for future research. Average nucleotide identity (ANI) analysis revealed seven cases of species-level sharing (ANI value > 95%) among hosts, including a putative novel Acetivibrionaceae species, suggesting possible cross-host transmission facilitated by sporulation. In all 63 genomes predicted to sporulate, we identified nine genes across sporulation steps. In addition, SHapley Additive exPlanations (SHAP) analysis indicated 16 consensus genes consistently contributing to predictions (namely lytH, cotP, spoIIIAG, spoIIR, spoVAD, gerC, yabP, yqfD, gerD, spoVAA, gpr, ytaF, gdh, ypeB, spoVID, and ymfJ), bringing biologically meaningful features across sporulation stages. By combining gene annotation with interpretable machine learning, SpoMAG provides a reproducible and accessible framework to infer sporulation potential in uncultured microbial taxa. This tool enhances targeted investigations into microbial survival strategies and supports research in microbiome ecology, probiotic discovery, food safety, and public health surveillance. SpoMAG is freely available as an R package and expands current capabilities for functional inference in metagenomic datasets.

Introduction

Sporulation is a key survival mechanism employed by members of the Firmicutes phylum (also known as Bacillota), allowing the formation of metabolically dormant spores highly resistant to extreme environmental stresses such as heat, desiccation, and low pH conditions (Eichenberger & Driks, 2014b). This evolutionary adaptation promotes bacterial persistence across different ecosystems (Swick, Koehler & Driks, 2016a; Li et al., 2024) and underlies significant practical applications in health and industry. The extensively characterized Bacilli and Clostridia classes include species of clinical relevance as well as strains used in biotechnology (Eichenberger & Driks, 2014a; Talukdar et al., 2015; Li et al., 2022; Guerrero & Gloria, 2023). While spores contribute to pathogen transmission (Paredes-Sabja & Sarker, 2009; Fischetti et al., 2019), they also enable beneficial functions, including use in probiotic formulations (Hong, Duc & Cutting, 2005; Elisashvili, Kachlishvili & Chikindas, 2019) and engineered delivery systems (Isticato, 2023; Saggese et al., 2023).

Despite ecological and biomedical relevance, most spore-forming bacterial diversity remains uncultured. This leaves the sporulation potential of many taxa unknown and limits our understanding of their capacity to form spores (Kapinusova, Lopez Marin & Uhlik, 2023; Machado et al., 2024). Although conventional methods require laboratory induction of sporulation in cultured isolates, metagenomics allows direct investigation of uncultivated species through the recovery of metagenome-assembled genomes (MAGs). This approach enables genome-based assessment of functional traits such as sporulation (Dias et al., 2025). This is particularly relevant for studying the human gut sporobiota, which accounts for about half of the gut microbiome (Tetz & Tetz, 2017; Egan et al., 2021).

Spore-forming gut bacteria contribute to host metabolism and examples include Clostridium, Blautia, and Ruminococcus species, which produce short-chain fatty acids such as butyrate, acetate, and propionate, regulating energy supply, cholesterol synthesis, and glucose homeostasis (Hou et al., 2022). They also modulate immune responses by inducing anti-inflammatory cytokines and strengthening gut barrier integrity (Dekeukeleire et al., 2025), influence serotonin biosynthesis and gastrointestinal motility (Yano et al., 2015), and facilitate microbial transmission and ecosystem stability (Swick, Koehler & Driks, 2016b; Choo et al., 2017). Determining sporulation potential in gut microbial communities is essential, as disturbances in spore-forming populations have been linked to dysbiosis and intestinal pathologies (Johanesen et al., 2015).

However, predicting sporulation potential from MAGs remains challenging, primarily because most candidate taxa cannot be cultured for phenotypic validation. Accurate prediction is critical for understanding host colonization dynamics, microbial transmission, and the maintenance of host homeostasis (Swick, Koehler & Driks, 2016a). Current sporulation prediction relies on conserved genomic markers, such as Spo0A or sporulation-specific sigma factors, identified in model organisms like Bacillus subtilis (Abecasis et al., 2013). Nevertheless, the generalizability of these markers to a broader taxonomic range (Galperin et al., 2022), especially when dealing with incomplete or poorly characterized MAGs, remains uncertain.

To address the challenge of inferring sporulation potential in bacterial MAGs, we developed SpoMAG, a supervised machine learning tool designed to predict sporulation potential directly from MAG annotations. SpoMAG enables high-throughput inference of sporulation capability in uncultivated bacteria, supporting applications in probiotic development, microbiome engineering, pathogen surveillance, and food safety monitoring. Importantly, it can also be applied to complete or draft genomes of pure strains, extending its usability beyond metagenomic datasets. To our knowledge, no other machine learning tool currently exists for predicting sporulation potential across MAGs or pure strain genomes. Thus, SpoMAG offers a critical resource for future studies into deciphering the ecological and biomedical roles of spore-forming bacteria across uncultivated microbial diversity.

Materials & Methods

Sample collection and bioinformatics processing

Samples were collected from the GUARANI One Health Brazilian Group Network (Lemos et al., 2022). Between February and April 2020, rectal swabs were obtained in triplicates from cattle (n = 30), swine (n = 15), poultry (n = 30), and human fecal samples (n = 32) (Lemos et al., 2022). The sampling encompassed five Brazilian geographic regions: Northern (Castanhal-Pará/PA), Southern (Blumenau-Santa Catarina/SC), Southeastern (Bragança Paulista-São Paulo/SP), Midwestern (Dourados-Mato Grosso do Sul/MS), and Northeastern (Fortaleza-Ceará/CE), as reported previously (Lemos et al., 2022; Machado et al., 2024).

DNA extraction, sequencing, and bioinformatics analyses followed established protocols described by Lemos et al. (2022). MAGs were evaluated for quality using the Minimum Information about a Metagenome-Assembled Genome (MIMAG) criteria, and only high-quality MAGs (>90% completeness, <5% contamination) (Bowers et al., 2017) were retained for downstream analyses. To assess species-level similarity between MAGs from different hosts, we applied the FastANI method to calculate Average Nucleotide Identity (ANI), by considering a fragment length of 1,020 bp (–fragLen 1020) (Hernández-Salmerón, Irani & Moreno-Hagelsieb, 2023).

Selection of sporulation-associated genes

A total of 160 genes associated with bacterial sporulation (Table S1) were selected based on their conserved presence across Firmicutes species and established functional roles in sporulation processes (Galperin et al., 2022). Genes primarily involved in housekeeping functions during vegetative growth were excluded, as they are ubiquitously present in vegetative cells and do not specifically contribute to sporulation (Galperin et al., 2022). The selected genes represent key steps in sporulation, including sporulation onset and checkpoints, Spo0A regulon, engulfment, sigma factor regulons (SigF, SigE, SigG, SigK), spore cortex formation, spore coat assembly, and germination.

Selection of publicly available genomes of sporulating and non-sporulating bacteria

For supervised machine learning model development, we compiled a dataset of 136 high-quality complete bacterial genomes from a published study (Galperin et al., 2022). The dataset was balanced, comprising 68 sporulating and 68 non-sporulating species (Table S2), to minimize bias during model training and evaluation. Spore-forming genomes included 34 Bacilli and 34 Clostridia species, representing the two major lineages known for sporulation within the Firmicutes phylum, thus ensuring phylogenetic diversity. The non-sporulating group included 30 Bacilli and 38 Clostridia genomes, selected to closely match phylogenetic diversity while maintaining a clear functional distinction (i.e., absence of sporulation capability).

Although certain sporulation-associated genes are partially conserved across Firmicutes, others are class-specific (e.g., Bacilli-exclusive regulators). Thus, including both Bacilli and Clostridia allowed the modeling approach to capture both universal and lineage-specific genetic signatures, enhancing generalizability across diverse Firmicutes species, even non-described-yet species from metagenomics studies.

All genomes were uniformly processed using the same annotation pipeline applied to the MAGs to ensure consistency. Open reading frames (ORFs) prediction was performed using Prodigal v.2.6.3 in single-genome mode (-p single) with a bacterial translation table (-g 1) (Hyatt et al., 2010). Functional annotations for each genome were performed using eggNOG-mapper v.2.0.8 (e-value ≤ 1e−5, identity > 60%, query/subject coverage > 60%) (Huerta-Cepas et al., 2017) and the Kyoto Encyclopedia of Genes and Genomes (KEGG) database (Kanehisa et al., 2025).

Data preparation for supervised machine learning models to predict sporulation probability

The target variable was encoded as a binary factor with the levels “Sporulating” and “Non-Sporulating”. To guarantee reproducibility, we implemented random stratified partitioning of the dataset into training (70%) and testing (30%) subsets using the “createDataPartition” function from the caret package in R (Kuhn, 2008), version 7.0.1. A binary presence/absence matrix of the sporulation genes was constructed prior to model training, representing fixed biological characteristics for each genome. Genomes included in the test set were completely withheld from the training process to prevent any form of data leakage. Given that only high-quality complete isolated genomes were used, missing values were treated as absences and encoded as zero, reflecting the assumption that unobserved genes were absent in the genomes.

Model training and tuning

We evaluated four supervised machine learning algorithms: Random Forest (RF), support vector machine (SVM), eXtreme Gradient Boosting (XGBoost), and neural network (NN). All models were implemented using caret (Kuhn, 2008) within a 10-fold cross-validation framework. Model performance was primarily assessed using the area under the receiver operative characteristic (ROC) curve (AUC).

The RF model was tuned by varying the number of variables randomly sampled at each split (mtry = 2, 3, or 4), while the number of trees (ntree) was fixed at 500. SVM was implemented with a radial basis function kernel. Hyperparameters were tuned over a grid of cost values (C = 0.25, 0.5, 1, 2, 4, 8, 16, 32, 64, and 128), while the kernel coefficient (sigma) was held constant at 0.0378. The XGBoost model was trained with learning rate (eta = 0.1), maximum tree depth (max_depth = 6), subsample ratio (subsample = 0.8), and column subsample by tree (colsample_bytree = 0.8). Early stopping was applied with a patience of 10 rounds, and the model was trained for a maximum of 100 rounds. Lastly, the NN was implemented as a feedforward multilayer perceptron using the nnet package, version 7.3-20 (Venables & Ripley, 2002). The architecture consisted of a single hidden layer, with the number of neurons tuned via repeated 10-fold cross-validation. Training was performed using backpropagation with a maximum of 200 iterations.

Development of the SpoMAG meta-classifier using an ensemble strategy

Based on their superior classification performance, RF and SVM algorithms were selected as base learners for the development of the SpoMAG, an ensemble classifier designed to predict probability of sporulation potential in MAGs. SpoMAG employs a stacked generalization strategy, where the probabilistic outputs of the RF and SVM models are used as input features for a meta-classifier. This meta-classifier was implemented as an RF model trained with 10-fold cross-validation (Ghasemieh et al., 2023), enhancing predictive robustness.

The meta-classifier’s hyperparameters, including the number of features considered at each split (mtry parameter), were optimized to improve classification performance. Both the base models and the ensemble were evaluated using standard metrics, including accuracy, recall, specificity, precision, F1-score, and AUC. To further assess predictive performance, we applied a non-parametric bootstrap procedure with 300 resampling iterations of the training set (Singh et al., 2021). Medians and 95% confidence intervals were calculated for all evaluation metrics, providing estimates while accounting for variability in the training data (Mokhtar, Yusof & Sapiri, 2023; Huang & Huang, 2023).

Before prediction, SpoMAG processes the functional annotation file of each MAG to identify genes based on gene names and KEGG Orthology (KO) identifiers. It then represents each MAG as a binary matrix encoding the presence or absence of 160 sporulation-associated genes and outputs a probabilistic estimate of sporulation potential for each genome.

Model interpretation and feature importance

To determine the most predictive genes for sporulation capability, we computed SHapley Additive exPlanations (SHAP) values (Lundberg & Lee, 2017) for both RF and SVM models using the iml package (Molnar, 2018), version 0.11.4, in R. The SHAP analysis measures the marginal contribution of each gene to individual predictions, offering an interpretable and consistent metric of feature importance.

Only genes identified as positive predictors in both RF and SVM models were selected for final biological interpretation. For visualization of feature importance, a bar plot was generated showing the mean absolute SHAP values of the contributing genes, arranged in descending order to highlight the most significant predictors of sporulation. The visualization was generated using ggplot2 (Wickham, 2016), version 3.5.1, and ggpubr (Kassambara, 2023), version 0.6.0, packages in R.

Ordination and multivariate dispersion analyses of sporulation profiles in Clostridia and Bacilli

To investigate variation in sporulation-associated gene profiles, we performed principal component analysis (PCA) using the prcomp function from the stats package (R Core Team, 2024), version 4.3.3. The analysis was based on binary matrices encoding the presence or absence of sporulation-associated genes across both reference genomes with known sporulation phenotypes and MAGs predicted as spore-forming by SpoMAG.

To assess potential differences in gene composition between the predicted sporulating and non-sporulating groups within the Clostridia and Bacilli classes, we performed a multivariate dispersion analysis using the betadisper function from the vegan package (Oksanen et al., 2024) (version 2.6.8). First, binary distance matrices were computed using the dist function (stats package), based on presence/absence profiles. The dispersion, i.e., within-group variability, was then calculated as the distance of each MAG to the centroid of its respective phenotypic group. Statistical significance of group dispersion differences was evaluated using 1,000 permutations and an analysis of variance (ANOVA) test at a 5% significance level.

Results

An overview of the main steps involved in model development and application is shown in Fig. 1. The training dataset comprised 136 bacterial genomes with experimentally validated sporulation phenotypes (Galperin et al., 2022), which were randomly partitioned into training (70%) and validation (30%) subsets. Four supervised learning algorithms were systematically evaluated: RF, SVM, XGBoost, and NN. Gene importance was evaluated using SHapley Additive exPlanations (SHAP) analysis for both RF and SVM models. Based on their performance, RF and SVM were selected as base learners for a stacked ensemble classifier named SpoMAG. The ensemble architecture used the predicted probabilities from each base model as input features for a meta-classifier, implemented as an RF model trained using 10-fold cross-validation.

Figure 1 Overview of SpoMAG model development and application workflow.

When applied to our target dataset, SpoMAG generated sporulation predictions for 63 genomes, including nine from the Bacilli class and 54 from Clostridia, confirming its taxonomic versatility. Genomic analysis revealed two significant gene sets: (1) nine universally conserved sporulation-associated genes across all predicted spore-formers (namely pth, yaaT, spoIIAB, spoIIIAE, spoIIIAD, ctpB, ftsW, spoVD, and lgt), and (2) sixteen classifier-informative genes identified through SHAP value analysis as consistently contributing to accurate phenotype prediction (namely lytH, cotP, spoIIIAG, spoIIR, spoVAD, gerC, yabP, yqfD, gerD, spoVAA, gpr, ytaF, gdh, ypeB, spoVID, and ymfJ).

The complexity of sporulation gene profiles challenges linear separation between sporulating and non-sporulating genomes

To explore whether the presence and absence of sporulation-associated genes alone could distinguish spore-forming from non-spore-forming organisms, we performed a PCA across the 136 Clostridia and Bacilli genomes, including both phenotypes (Fig. 2). The first principal component (PC1) accounted for 41.99% of the total variance, followed by PC2 with 14.48%. While PCA captured discernible clustering patterns, we observed significant phenotypic overlap between spore-forming and non-spore-forming genomes in both phylogenetic classes. This substantial overlap demonstrates that dimensionality reduction of binary gene presence-absence data alone provides insufficient resolution for reliable sporulation phenotype prediction.

Figure 2 Principal component analysis of sporulating and asporogenic Clostridia and Bacilli genomes, based on binary presence/absence of sporulation-associated genes.

Each geometric shape represents an individual genome.

Comparison of supervised machine learning models for classifying genomes as sporulating or non-sporulating

Among the four supervised classification models evaluated, RF and SVM exhibited the highest median performance in distinguishing sporulating from non-sporulating bacterial genomes (Table 1). The RF model showed the highest median accuracy (87.5%) and specificity (85.7%), reflecting robust generalization and a low false-positive rate, with confidence intervals indicating consistent predictions. Meanwhile, SVM exhibited the highest median recall (95.2%), indicating a high sensitivity for correctly identifying spore-forming genomes, albeit with lower specificity (75%). Both models reached comparable median AUC values (RF 89.7%, SVM 93%), reflecting their discriminative capabilities.

Table 1 Performance metrics of supervised machine learning models for classifying genomes as sporulating or non-sporulating.

Values represent the median and the 95% confidence interval.

Machine learning models	Accuracy (%)	AUC (%)	Recall (%)	Specificity (%)	Precision (%)	F1-score (%)	
Random forest	87.5
(77.5–97.5)	89.7
(80.5–97.2)	90.5
(75–100)	85.7
(68–100)	86.4
(68.6–100)	88
(75–97.3)	
Support vector machine	85
(72.5–95)	93
(83.1–99)	95.2
(83.3–100)	75
(55–93.3)	80
(61.9–95)	86.5
(73.7–95.7)	
XGBoost	82.6
(72.5–92.5)	93.3
(85.8–99)	95
(85.7–100)	70.2
(52.6–86.4)	76.2
(61.5–89.5)	84.2
(73.7–93)	
Neural network	75
(60–87.5)	91.1
(80.3–97.7)	85.7
(68.2–100)	65.2
(43.5–85.7)	71.4
(52.2–88.5)	77.6
(61.1–89.4)	
Stacking ensemble (SpoMAG)	87.5
(77.5–97.5)	92.2
(81.1–99.7)	90.5
(75–100)	85.7
(68.2–100)	86.4
(69.6–100)	88.2
(75–97.3)	

To leverage the complementary strengths of RF and SVM, we developed SpoMAG, a stacking ensemble classifier that integrates these two base learners. SpoMAG preserved the high median accuracy of RF (87.5%) and recall (90.5%), also achieving the highest median F1-score (88.2%) among all models. Importantly, it showed AUC of 92.2%, confirming that the ensemble approach enhanced predictive balance and stability without compromising classification performance.

SpoMAG demonstrates high specificity in predicting the absence of sporulation in non-Firmicutes genomes

To evaluate the specificity of SpoMAG, we applied the model to a dataset of 496 high-quality MAGs from bacterial phyla outside Firmicutes, including genomes derived from cattle (n = 136), poultry (n = 126), human (n = 105), and swine (n = 129). As expected, SpoMAG assigned a predicted sporulation probability of zero to all MAGs (Fig. 3, Table S3), demonstrating SpoMAG’s high specificity and its ability to correctly distinguish non-sporulating genomes, even in complex metagenomic datasets. This performance underscores SpoMAG’s robustness and affirms its applicability for large-scale genome annotation across heterogeneous microbiomes.

Figure 3 Probability density plots of predicted sporulation likelihoods for non-Firmicutes MAGs from (A) cattle, (B) poultry, (C) humans, and (D) swine.

Sporulation potential across Bacilli-class MAGs reveals order-specific patterns

A total of 51 high-quality MAGs from the Bacilli class were recovered across cattle (n = 10 MAGs), poultry (n = 15 MAGs), human (n = 20 MAGs), and swine (n = 6 MAGs). SpoMAG predictions revealed distinct order-level sporulation patterns consistent with established phenotypic characteristics of Bacilli lineages (Fig. 4, Table S4).

Figure 4 Predicted sporulation probabilities for Bacilli-class MAGs across hosts.

Percentages represent average probability scores per host.

In cattle microbiota, one MAG from the order Paenibacillales (family Paenibacillaceae, genus Cohnella) was predicted to be spore-forming with 99% probability (Fig. 4), consistent with the established sporulation capability of this order. MAGs from other orders, including RF39, ML615J-28, Izemoplasmatales, Erysipelotrichales, and Acholeplasmatales, were all assigned 0% sporulation probability, corroborating their non-sporulating status (Table S4).

Poultry-derived MAGs exhibited a similar trend, with four MAGs from Paenibacillales (family Paenibacillaceae with three from Paenibacillus and one from Cohnella) predicted as spore-formers with 99% probability (Fig. 4). Two Bacillales MAGs (family Planococcaceae, genus Rummeliibacillus) showed an average sporulation probability of 81.1%. All remaining MAGs from Erysipelotrichales and Lactobacillales scored 0%.

Among human-derived MAGs, two genomes were predicted as sporulating: one from Bacillales (Planococcaceae, genus Ureibacillus) and another from DSM-1321 family (genus Pradoshia), both with average sporulation probability of 93%. All other human MAGs, including those from Erysipelotrichales and Lactobacillales, were predicted as non-spore-forming.

Notably, swine samples lacked Bacillales and Paenibacillales representatives. All six swine MAGs belonged to orders traditionally considered non-sporulating (e.g., Acholeplasmatales, Erysipelotrichales, Lactobacillales, and RFN20), and all were assigned a 0% probability by SpoMAG.

Analysis of gene content further supported SpoMAG’s predictions, with MAGs from Paenibacillales and Bacillales containing a higher number of sporulation-associated genes compared to other Bacilli orders (Fig. 5, Table S5). All MAGs from other Bacilli orders scored 0% of sporulation probability, reinforcing the tool’s biological consistency and specificity.

Figure 5 Comparison of sporulation-associated gene abundance between spore-forming (Paenibacillales and Bacillales) and non-spore-forming Bacilli MAGs.

At the end of the order names, SPO indicates a predicted spore-former, and NONSPO indicates a predicted non-spore-former.

To explore within-group variability in gene profiles, we performed a multivariate dispersion analysis (Betadisper) in the predicted spore-formers and non-spore-formers from Bacilli. The analysis revealed significant differences in gene profile dispersions (F = 12.62, p =0.0007), indicating distinct within-group variability in gene composition.

Sporulation potential across Clostridia-class MAGs reveals order-specific patterns

We recovered 262 high-quality MAGs from the Clostridia class, distributed across cattle (n = 53 MAGs), poultry (n = 58 MAGs), humans (n = 95 MAGs), and swine (n = 56 MAGs). SpoMAG predictions revealed consistent and order-specific patterns in predicted sporulation potential (Fig. 6, Table S6).

Figure 6 Predicted sporulation probabilities for Clostridia-class MAGs across hosts.

Percentages represent the average probability scores per host.

In human samples, seven MAGs from the order Acetivibrionales were classified as spore-forming, with an average predicted probability of 98.5%. All belonged to the family Acetivibrionaceae, including two from the genus Ruminiclostridium and five unclassified at the genus level. One MAG from the same order, classified under (DSM-8532 family) was predicted to be non-sporulating (0% probability). Within Christensenellales, only one of 19 MAGs was predicted as a spore-former (Christensenellaceae, unknown genus/species). Among the remaining, mostly from poorly characterized families and the Borkfalkiaceae, were classified as non-sporulating. All Clostridiales MAGs (n = 3) were predicted to be spore-forming, with three from the Clostridiaceae family (one identified as Clostridium and two unclassified at the genus level). In contrast, all 24 MAGs from Lachnospirales were classified as non-sporulating, including genera within Lachnospiraceae such as Herbinix, Lachnoclostridium, Butyrivibrio, UBA4285, RUG115, Blautia, Coprococcus, Enterocloster, KLE1615, CAG-194, Faecalimonas, Anaerostipes, Mediterraneibacter, CAG-127, Dorea, Acetatifactor, CAG-603, Bariatricus, and Anaerocolumna, all lacking species-level classification.

The order Monoglobales included one MAG predicted as a spore-former (family UBA1381, genus CAG-41) and one as a non-spore-former (family Monoglobaceae, genus Monoglobus). Among 34 MAGs from Oscillospirales, one was predicted as spore-forming (family Ruminococcaceae, genus UBA3818), potentially representing a novel spore-forming species. All MAGs from Peptostreptococcales, Tissierellales, and UMGS1883 were predicted to be non-spore-forming.

In swine, SpoMAG predicted sporulation in eight Acetivibrionales MAGs, including Acetivibrionaceae (n = 7) and DSM-8532 (n = 1), all unclassified at the species level. Additionally, three Christensenellales MAGs were predicted as spore-formers, including one from Christensenellaceae and two from poorly defined families (CAG-74 and SZUA-584). Two Clostridiales MAGs (Clostridiaceae, genus Clostridium) and one Oscillospirales MAG (Ruminococcaceae, unknown genus and species) were also identified as spore-formers. All MAGs from DUPQ01, Lachnospirales, Peptostreptococcales, Saccharofermentanales, and Tissierellales were predicted as non-sporulating.

In cattle, SpoMAG identified six spore-forming MAGs from Acetivibrionales (three from Acetivibrionaceae, three from DSM-8532), two from Christensenellales (families Christensenellaceae and SZUA-584), and three from Clostridiales (two Clostridiaceae, one Caloramatoraceae, unknown genus). Notably, two MAGs from Lachnospirales (Herbinix genus, family Lachnospiraceae) were also predicted to be spore-forming. One MAG from Peptostreptococcales (Anaerovoracaceae, genus UBA7709) was classified as a spore-former.

In poultry, SpoMAG predicted all seven Acetivibrionales MAGs as spore-formers (six from Acetivibrionaceae and one from DSM-8532). All MAGs from Clostridiales (n = 2) were also classified as spore-forming, along with one Christensenellales (Christensenellaceae, unknown species) and one Peptostreptococcales (Natronincolaceae, genus Alkaliphilus). Two MAGs from Lachnospirales (genus Herbinix) were also predicted as spore-formers.

Gene presence/absence patterns are insufficient to distinguish sporulation in Clostridia MAGs

We next investigated whether presence/absence patterns of sporulation-associated genes could distinguish between predicted spore-forming and non-spore-forming MAGs. Unlike Bacilli, binary gene profiles were insufficient to reliably separate sporulation capability in Clostridia (Fig. 7, Table S7). Although some differences in gene content were observed, there was substantial overlap between phenotypic groups, suggesting that gene presence alone may indicate a latent potential to sporulate rather than active functionality. Importantly, the presence of sporulation-associated genes reflects genetic potential, not active gene expression at the time of sampling.

Figure 7 Comparison of sporulation-associated gene content between predicted spore-forming and non-spore-forming Clostridia MAGs.

At the end of the order names, SPO indicates a predicted spore-former, and NONSPO indicates a predicted non-spore-former.

To evaluate gene content variation in more detail in the Clostridia class, we performed PCA on sporulation gene profiles (Fig. 8). In cattle (Fig. 8A), poultry (Fig. 8B), human (Fig. 8C), and swine (Fig. 8D), there was considerable overlap between predicted spore-forming and non-spore-forming MAGs. Although distinct ellipses were generated for each group, indicating some variation, no clear separation was observed in the multivariate space, further demonstrating that unsupervised ordination methods still insufficient to resolve sporulation status even for the predicted Clostridia MAGs.

Figure 8 Principal Component Analysis of sporulation-associated gene profiles in Clostridia MAGs from (A) cattle, (B) poultry, (C) human, and (D) swine.

Each point represents a MAG, colored by predicted sporulation phenotype (sporulating vs. non-sporulating). The ellipses represent 95% confidence intervals around the group centroids.

We then assessed within-group dispersion using betadisper analysis. Statistically significant differences in dispersion were observed in the spore-forming and non-spore-forming MAGs for all hosts: cattle (F = 7.55, p = 0.0083), swine (F = 43.77, p < 0.001), human (F = 14.19, p = 3 ×10−4), and poultry (F = 13.42, p = 6 ×10−4), indicating that the two groups differ in internal variability of sporulation gene content, suggesting distinct levels of genomic variability related to sporulation potential within each group.

SpoMAG reveals potentially sporulating Firmicutes species shared across hosts

Our comprehensive pairwise average nucleotide identity (ANI) analysis of high-quality Firmicutes MAGs revealed distinct distribution patterns among genomes predicted to be sporulating (Table S8). Most MAGs, both predicted sporulating and non-sporulating, exhibited ANI values below the 95% species threshold. A notable proportion clustered at 74% ANI, indicating considerable species-level diversity among the recovered genomes (Fig. 9A).

Figure 9 Average nucleotide identity (ANI) values for MAGs from the four hosts.

(A) Frequency of pairwise genome ANI with a focus on >99% ANI values. (B) Species sharing at the species level across poultry, swine, human, and cattle hosts. Taxonomic levels were represented by standard prefixes: o_ (order), f_ (family), g_ (genus), and s_ (species). Icon source credit: BioRender.

Despite this diversity, SpoMAG-based predictions revealed seven cases of species-level sharing (ANI value > 95%) among spore-forming MAGs recovered from different hosts (Fig. 9A). These included two shared species between poultry and swine, two between poultry and humans, one between swine and humans, and two between swine and cattle (Fig. 9B). Notably, a species from the Acetivibrionaceae family exhibited ∼99% ANI across poultry, swine, and human hosts.

MAGs exhibit distinct sporulation gene profiles at the Class and Order levels across hosts

To explore patterns in sporulation gene content across hosts, we performed PCA on the binary presence/absence matrices of sporulation-associated genes in MAGs predicted to be spore-forming by SpoMAG. The ordination revealed class- and order-specific clustering, as well as host-associated trends.

Human-derived MAGs (Fig. S1, Table S9) had the first two principal components explaining 42.33% of the total variance. The two Bacilli MAGs were clearly separated from Clostridia. Within Clostridia, seven Acetivibrionales MAGs (all from Acetivibrionaceae family) clustered tightly, suggesting conserved sporulation gene profiles. Notably, two of these MAGs are assigned to Ruminiclostridium genus (bin.488_human and bin.208_human).

In swine MAGs (Fig. S2, Table S9), only Clostridia were predicted as spore-formers. The first two components explained 40.57% of the variance and revealed two distinct clusters: one composed of eight Acetivibrionales MAGs, and another of three Christensenellales MAGs. The remaining MAGs were mixed into a cluster of two MAGs from Clostridiales and one from Oscillospirales.

Cattle-derived MAGs (Fig. S3, Table S9) had PC1 and PC2 explaining 34.88% of the variance. Six Acetivibrionales MAGs formed a cluster, adjacent to two Lachnospirales MAGs, indicating partial similarity in gene profiles. A single Bacilli MAG (order Paenibacillales) appeared well separated from Clostridia. A separate group included MAGs from Christensenellales (n = 2), Clostridiales (n = 3), and Peptostreptococcales (n = 1).

In poultry-derived MAGs (Fig. S4, Table S9), the first two components explained 49.20% of the variance. Two Bacilli MAGs and four from Paenibacillales formed distinct clusters outside the main Clostridia group. Within Clostridia, the mixed cluster included MAGs from Acetivibrionales (n = 7), Christensenellales (n = 1), Clostridiales (n = 2), Lachnospirales (n = 2), and Peptostreptococcales (n = 1), again reflecting compositional variability.

When combining all 63 MAGs predicted to be spore-formers, the PCA revealed a separation between Bacilli and Clostridia, with the first two components explaining 32.21% of the variance (Fig. 10, Table S9). Five Paenibacillales MAGs grouped together, including one MAG from Bacillales. A well-defined cluster of 28 Acetivibrionales MAGs confirmed the conservation of sporulation gene profiles within this order, even among hosts. The remaining Clostridia MAGs formed a broader group, representing the orders Christensenellales (n = 7), Clostridiales (n = 10), Lachnospirales (n = 4), Monoglobales (n = 1), Oscillospirales (n = 2), and Peptostreptococcales (n = 2).

Figure 10 Principal component analysis of MAGs, from the four hosts, predicted to sporulate by SpoMAG.

Consistent genes present across the 63 predicted sporulating MAGs

We identified nine genes consistently present in all 63 MAGs predicted as spore-formers by SpoMAG (Fig. 11). These include: pth, yaaT, spoIIAB, spoIIIAE, spoIIIAD, ctpB, ftsW, spoVD, and lgt. Although an additional 13 genes were found in 62 of the 63 genomes, we focused on those shared across all genomes to ensure consistency in gene presence.

Figure 11 Distribution of distinct sporulation-associated genes in the 63 MAGs.

Nine genes are consistently present in all MAGs predicted to sporulate (in red). SOAC, sporulation onset and checkpoints.

Functionally, pth and yaaT are associated with early-stage regulation, while spoIIAB, spoIIIAE, and spoIIIAD contribute to initiation and engulfment; ctpB and ftsW are linked to the SigE regulon; spoVD is involved in spore cortex formation; and lgt plays a role in spore germination.

SHAP analysis identified 16 genes consistently contributing to sporulation prediction in both Random Forest and Support Vector Machine models (Fig. 12). These include: lytH, cotP, spoIIIAG, spoIIR, spoVAD, gerC, yabP, yqfD, gerD, spoVAA, gpr, ytaF, gdh, ypeB, spoVID, and ymfJ. All genes exhibited positive SHAP values, indicating their influence on model classification decisions.

Figure 12 Identification of 16 genes with positive SHAP values impacting SpoMAG’s performance.

The genes lytH, yabP, and yqfD are associated with the spore cortex; cotP and spoVID are involved in the spore coat; spoIIIAG in engulfment; spoIIR, ymfJ are part of the SigF regulon; spoVAD, spoVAA are related to the SigG regulon; ytaF in SigE regulon; and gerC, gerD, gpr, gdh, ypeB are involved in germination.

Discussion

Here, we present SpoMAG, an R-based machine learning tool for predicting the sporulation potential of MAGs from uncultivated Firmicutes. Unlike traditional approaches relying on conserved marker genes or gene counts, SpoMAG combines genome annotations with an ensemble learning strategy to capture complex presence/absence patterns of sporulation-associated genes. The model was trained on genome annotations from experimentally validated sporulation phenotypes and provides probabilistic predictions of sporulation capacity. This advances functional inference for uncultured bacterial species, particularly relevant given that >70% of gut microbiome members lack cultivation-based characterization (Almeida et al., 2021).

We first examined whether sporulation-associated gene presence or absence could distinguish spore-forming from non-spore-forming genomes in Bacilli and Clostridia using PCA. Notably, substantial overlap was observed between the two phenotypic groups in both classes, indicating that binary gene profiles alone lack the resolution needed for accurate classification. Furthermore, genes previously proposed as sporulation markers (Onyenwoke et al., 2004) have since been identified in non-spore-formers (Galperin et al., 2012), underscoring the limitations of relying on individual genes as definitive indicators. This observed complexity supports the necessity for complementary strategies, such as machine learning, to capture more complex patterns and improve the prediction of sporulation potential.

To overcome these limitations, we applied a stacking ensemble integrating RF and SVM. The ensemble retained RF’s accuracy (87.5%) and recall (90.5%), while achieving the highest F1-score (88.2%), reflecting better balance between sensitivity and precision. SpoMAG also maintained a high AUC (92.2%), suggesting enhanced predictive stability without compromising performance, a trend consistent with a prior study (Lin et al., 2022). Collectively, these results validate the effectiveness of ensemble learning in capturing complex patterns in biological datasets (Nagi & Bhattacharyya, 2013; Zolfaghari et al., 2023).

To evaluate SpoMAG’s specificity, we applied it to 496 high-quality MAGs from bacterial phyla outside Firmicutes. As expected, all were assigned a sporulation probability of zero, consistent with the understanding that endospore formation is largely restricted to Firmicutes (De Hoon, Eichenberger & Vitkup, 2010; Galperin, 2013; Remize & De Santis, 2025), and underscores SpoMAG’s robustness in avoiding false positives in distantly related bacterial species.

Application to Bacilli MAGs confirmed SpoMAG’s ability to recover known spore-formers. All MAGs from Bacillales and Paenibacillales, two orders known for spore formation (Galperin, 2016; Zander, Schmid & Kabisch, 2024; Zhang et al., 2025), were assigned high sporulation probability scores. In contrast, MAGs from non-spore-forming orders, such as Lactobacillales (Galperin, 2013), were consistently predicted as non-spore-forming. Comparative gene content analysis further supported these predictions, with Bacillales and Paenibacillales MAGs containing more sporulation-associated genes than non-sporulating Bacilli orders. However, we also observed variability in gene counts within both predicted spore-formers and non-spore-formers, reinforcing that the specific combination of these genes, rather than just their number in a given genome, are critical for sporulation possibility.

We expanded our analysis to Clostridia, a class characterized by heterogeneous sporulation phenotypes, where both spore-forming and non-spore-forming species frequently co-occur within the same taxonomic orders (Yutin & Galperin, 2013; Liu et al., 2024; Weis et al., 2024). SpoMAG revealed distinct host-associated and order-specific sporulation potential patterns among the analyzed MAGs. These findings demonstrate SpoMAG’s ability to discern functional potential variation across phylogenetically and ecologically diverse genomes.

The consistent prediction of sporulation across all Acetivibrionaceae members from all hosts is particularly significant, given this family’s established role in nutrient scavenging (Dias et al., 2025). SpoMAG’s identification of Ruminiclostridium MAGs in human samples corroborates its predictive accuracy, as this genus is extensively documented for both sporulation and carbohydrate fermentation (Yutin & Galperin, 2013; Wu & Cheng, 2021). These results may facilitate targeted cultivation strategies and enhance our comprehension of host-microbe interactions. Similarly, SpoMAG’s identification of spore-forming potential in Christensenellaceae MAGs aligns with recent experimental evidence of endospore formation in cultured strains from human feces (Sun et al., 2024), validating the tool’s capacity to detect spore-forming potential traits in poorly characterized taxa.

All Clostridiales MAGs were classified as spore-formers by SpoMAG, consistent with their established sporulating capacity (Paredes-Sabja, Setlow & Sarker, 2011; Fischetti et al., 2019). Conversely, human-derived MAGs from Lachnospirales were predicted as non-sporulating, in agreement with previous findings that members of this order generally lack sporulation capability (Abdullah et al., 2023). However, sporadic occurrences of sporulation have been observed in certain strains, supporting the recognized phenotypic heterogeneity within Clostridia (Haas & Blanchard, 2017).

SpoMAG’s divergence prediction of sporulation potential in cattle- versus poultry-associated Herbinix MAGs emphasizes important host specific differences that may reflect niche adaptation (Browne et al., 2021). While Herbinix is associated with fiber degradation (Koeck et al., 2015; Koeck, Hahnke & Zverlov, 2016), the maintenance of sporulation capacity in cattle-associated MAGs may provide an ecological advantage in a gut environment characterized by long retention times and a fiber-rich diet (Osorio-Doblado et al., 2023). In poultry, sporulation in Herbinix could instead favor transmission, as the rapid intestinal transit and constant exposure to the environment might select for spore formation as a survival and dispersal strategy (Rougière & Carré, 2010; Clavijo & Flórez, 2018). Similar heterogeneity was observed in Peptostreptococcales, where sporulation potential was restricted to cattle and poultry MAGs suggesting potential ecological specialization or evolutionary divergence among host-associated lineages. Such lineage-specific differences may indicate secondary loss of sporulation in non-ruminant/non-avian hosts or, alternatively, the retention of sporulation as a host-dependent trait shaped by selective pressures.

SpoMAG’s prediction of sporulation potential in understudied groups as the families UBA1381 (order Monoglobales) and Ruminococcaceae (genus UBA3818) suggests that these MAGs may represent previously unrecognized spore-formers worthy of targeted cultivation efforts. Their presence in both human and swine samples supports their ecological relevance and may contribute to the evolutionary role of sporulation within Clostridia, particularly in orders that remain poorly characterized and uncultivated. These findings illustrate how machine learning approaches can complement culture-based methods to reveal overlooked microbial functional traits, particularly in challenging-to-study gut bacteria.

Unlike in Bacilli, where sporulation gene presence/absence correlated with the predicted phenotype, Clostridia MAGs exhibited partial conservation of sporulation genes even among non-spore-formers. This is consistent with prior reports of conserved sporulation-related genes in Clostridia regardless of functional phenotype (Talukdar et al., 2015; Connor et al., 2019). Despite this, we detected significant dispersion differences between predicted spore-forming and non-spore-forming groups across all hosts, indicating that while individual gene presence is unreliable, the complete sporulation network complements the phenotypic sporulation potential. These findings also highlight why machine learning approaches such as SpoMAG can outperform gene-counting methods. By evaluating coordinated patterns across multiple sporulation determinants, SpoMAG effectively infers sporulation probability even in the presence of partial or incomplete gene sets, supporting the importance of multivariate strategies for investigating complex traits within taxonomically diverse groups such as Clostridia. Moreover, the within-group clustering observed in Clostridia PCA plots suggests that true spore-formers can share conserved genomic sporulation signatures, reinforcing SpoMAG’s ability to detect functional cohesion despite incomplete genome recovery.

ANI analysis revealed seven cases of species-level sharing (ANI > 95%) among predicted spore-forming MAGs across hosts, including a potentially novel Acetivibrionaceae species found in poultry, swine, and human microbiomes. While this broad host distribution indicates that certain spore-formers may occur in multiple hosts, possibly facilitated by their sporulation ability (Swick, Koehler & Driks, 2016b), it does not provide direct evidence of cross-species transmission, as ecological and temporal data are lacking. However, it is important to note that the presence of genetically identical species in phylogenetically distant hosts challenges traditional views of host specificity in gut microbiota and raises the possibility of shared environmental reservoirs or dietary transmission routes. Given that sporulation confers high stability and resilience to environmental stresses, broadly distributed spore-forming lineages may constitute suitable targets for probiotic development. In this context, spore formation has been recognized as a key trait for ensuring product stability and successful gut colonization (Hong, Duc & Cutting, 2005; Bader, Albin & Stahl, 2012; Ahire, Kashikar & Madempudi, 2021). These findings could contribute to future research into microbial ecology, zoonotic risk assessment, probiotics development and the biotechnological exploitation of resilient, broadly distributed bacterial species.

The PCA of sporulation gene presence/absence profiles demonstrated hierarchical conservation patterns, revealing clear separation between Bacilli and Clostridia that reflects their divergent evolutionary histories of sporulation gene acquisition and regulation (Talukdar et al., 2015). Within Clostridia, the 28 Acetivibrionaceae MAGs formed a tightly clustered group despite being recovered from different hosts, indicating highly conserved sporulation gene content within this family. This clustering mirrors the ANI-based similarity among these genomes, reinforcing that both genomic architecture and sporulation-associated gene content are conserved across host environments.

The combination of ANI and PCA approaches provides complementary information, while ANI analysis identifies potential cross-host occurrence of specific spore-forming species, PCA reveals conserved functional patterns associated with sporulation capability across phylogenetic groups. This is particularly relevant for clinically important pathogens like Clostridioides difficile, which demonstrate both high transmissibility and zoonotic potential (Deakin et al., 2012; Tsai et al., 2016; Tsai et al., 2021; Knight & Riley, 2019). Our detection of genetically identical species in human and animal microbiomes reinforces these epidemiological concerns and highlights the utility of SpoMAG for public health surveillance.

Moreover, current spore mitigation strategies, including high-pressure processing to induce germination followed inactivation (Delbrück et al., 2021), face challenges from superdormant spore subpopulations resistant to germination, posing challenges to spore inactivation (Zhang & Mathys, 2018; Delbrück et al., 2021; Delbrück et al., 2022). Thus, SpoMAG’s ability to characterize sporulation traits could aid in identifying bacterial species possessing such resistance traits, thereby contributing to novel food safety interventions.

We identified a subset of nine genes consistently present in all 63 MAGs classified as spore-formers, as well as 16 consensus genes that significantly contributed to classification performance in both the RF and SVM models. These two gene sets likely represent important components of the sporulation machinery. Functionally, they span multiple stages of the sporulation process, from early regulatory checkpoints to forespore engulfment, cortex synthesis, and spore germination, indicating their potential role in uncultivated Firmicutes. The consistent presence of these genes and the contribution to predictive modeling emphasize promising candidates for future experimental validation and potential targets for antimicrobial strategies (Galperin et al., 2022).

Contamination by spore-forming bacteria is a major concern in the food industry, particularly in dairy products such as milk powder (Ruis, Fröder & Perez, 2025). Spores are highly resistant and capable of surviving heat treatments, enabling them to germinate post-processing and potentially lead to spoilage (Navaneethan & Effarizah, 2023). Current detection methods, including culture-based techniques and 16S rRNA sequencing, are limited by low sensitivity or dependence on cultivation (Li et al., 2018; Murphy et al., 2019). SpoMAG can complement these approaches by predicting sporulation potential. When integrated with experimental validation in a hybrid framework, it also facilitates iterative improvement of ML-based classification models, ultimately enhancing SpoMAG’s robustness and applicability to food safety monitoring.

SpoMAG demonstrates multidisciplinary utility through distinct applications across microbial sciences. In ecological studies, it facilitates systematic investigation of sporulation dynamics in natural populations, particularly within host-associated microbiomes where sporulation influences persistence, transmission, and community stability (Swick, Koehler & Driks, 2016b; Browne et al., 2021). For clinical applications, SpoMAG enables identification of spore-forming commensals or opportunistic pathogens capable of resisting environmental stress or antimicrobial treatment. In biotechnology, SpoMAG supports targeted selection of industrial-relevant strains for probiotic development, bioremediation applications, and fermentation processes, where sporulation impacts process efficiency and product stability.

Notably, bacterial endospores have been identified in bovine feed stocks and manure deposits, which they can contaminate milk, particularly when bedding materials create microenvironments conducive to sporulation (Vissers et al., 2007). These epidemiological observations underscore the necessity of functional prediction tools for dairy safety monitoring. The systematic identification of sporulation capacity in phylogenetically diverse, uncultivated lineages represents a critical research frontier in microbial physiology. SpoMAG addresses this challenge by providing a computational framework that translates metagenomic signatures into functional sporulation potential.

Future investigations should prioritize experimental validation of SpoMAG predictions through targeted isolation of putative spore-formers, particularly within undercharacterized Clostridia lineages. Furthermore, SpoMAG’s application to environmental metagenomes, including agricultural, food processing, and soil studies, may reveal previously unrecognized ecological dimensions of sporulation beyond host-associated ecosystems.

Conclusions

This study presents SpoMAG, a supervised machine learning tool that predicts sporulation potential in MAGs from uncultivated Firmicutes. By combining gene annotations with ensemble learning, SpoMAG achieved high-accuracy discrimination between spore-forming from non-spore-forming genomes and revealed biologically relevant gene patterns across 63 MAGs derived from vertebrate hosts. The tool enabled functional inference in uncultured bacteria species, overcoming the limitations of marker-based or purely taxonomic approaches. While experimental validation remains a future direction, particularly for novel Clostridia candidates, SpoMAG offers a robust and reproducible effort for investigating bacterial survival strategies in host-associated and environmental microbiomes.

Supplemental Information

Supplemental Information 1 Principal component analysis of human-derived MAGs predicted to sporulate by SpoMAG

Supplemental Information 2 Principal component analysis of swine-derived MAGs predicted to sporulate by SpoMAG

Supplemental Information 3 Principal component analysis of cattle-derived MAGs predicted to sporulate by SpoMAG

Supplemental Information 4 Principal component analysis of poultry-derived MAGs predicted to sporulate by SpoMAG

Supplemental Information 5 List of the 160 genes associated with bacterial sporulation

Supplemental Information 6 Dataset of 136 bacterial genomes with known sporulation phenotype

Supplemental Information 7 SpoMAG results in MAGs classified outside the Firmicutes phylum

Supplemental Information 8 SpoMAG results in MAGs from Bacilli

Supplemental Information 9 Gene content for MAGs from Bacilli

Supplemental Information 10 SpoMAG results in MAGs from Clostridia

Supplemental Information 11 Gene content for MAGs from Clostridia

Supplemental Information 12 Average Nucleotide Identity results

Supplemental Information 13 MAGs from the four hosts used for the PCA analyses

We thank all researchers from the seven participating centers belonging to the GUARANI (Grupo Brasileiro de Saúde Única) One Health Network for their commitment and hard work: Regional University of Blumenau (FURB)—Alessandro Conrado de Oliveira Silveira and Eleine Kuroki Anzai; Instituto Evandro Chagas (IEC)—Cintya de Oliveira Souza, Danielle Murici Brasiliense, Márcia de Nazaré Miranda Bahia, William Alencar de Oliveira Lima; Federal University of Ceará (UFC)—Débora de Souza Collares Maia Castelo-Branco and Glaucia Morgana de Melo Guedes; University São Francisco (USF)—Lúcio Fábio Caldas Ferraz and Walter Aparecido Pimentel Monteiro; Universidade Federal de São Paulo (UNIFESP)—Ana Cristina Gales, Carlos Roberto Vieira Kiffer, Fernanda Fernandes Santos, Francisco Ozório Bessa-Neto, Tiago Barcelos Valiatti, Raissa Fidelis Baeta Neves, Ramon Giovani Brandão da Silva, Rodrigo Cayô, and Ruanita Veiga Queiroz Apolinário.

Additional Information and Declarations

Competing Interests

Author Contributions

Data Availability

Ana Tereza Ribeiro Vasconcelos is an Academic Editor for PeerJ.

Douglas Terra Machado conceived and designed the experiments, performed the experiments, analyzed the data, prepared figures and/or tables, authored or reviewed drafts of the article, and approved the final draft.

Otávio José Bernardes Brustolini conceived and designed the experiments, authored or reviewed drafts of the article, and approved the final draft.

Ellen dos Santos Corrêa performed the experiments, authored or reviewed drafts of the article, and approved the final draft.

Ana Tereza Ribeiro Vasconcelos conceived and designed the experiments, authored or reviewed drafts of the article, coordinated all steps of the work, and approved the final draft.

The following information was supplied regarding data availability:

SpoMAG and the R package are available at GitHub: https://github.com/labinfo-lncc-br/SpoMAG.

The Supplemental tables are available in the Supplemental Files and at Zenodo: Terra Machado, D. (2025). Supplementary tables - 01:09 [Data set]. Zenodo. https://doi.org/10.5281/zenodo.15715511.

The R package is available at: https://cran.r-project.org/web/packages/SpoMAG.

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
