# Peer review of "Prediction of sporulating Firmicutes from uncultured gut microbiota using SpoMAG, an ensemble learning tool"

_PeerJ, doi:10.7717/peerj.20232_

## Round 0.1 · original submission · Major Revisions

Please revise the manuscript based on the reviewers' comments accordingly.

**Language Note:** The review process has identified that the English language must be improved. PeerJ can provide language editing services - please contact us at [email protected] for pricing (be sure to provide your manuscript number and title). Alternatively, you should make your own arrangements to improve the language quality and provide details in your response letter. – PeerJ Staff

Reviewer 1 ·

Basic reporting

Gene-panel justification thin – the 160 markers are said to be “conserved” but no selection flow or literature table is provided; please supply rationale and exclusion criteria.

No confidence intervals – AUC, F1 and other metrics are point values; 95 % CI or permutation test would strengthen statistical rigor.

Risk of class leakage – MAGs and reference genomes were annotated by the same pipeline; please clarify whether KO identifiers from test appear during feature engineering stage of training.

Over-interpretation of ANI sharing – claiming cross-host transmission from seven shared species is premature without ecological metadata or temporal sampling.

Figure resolution uneven – Fig 6 and Fig 7 fonts blur when zoomed, hard for readers; recommend higher dpi and vector export.

Raw data accessibility – raw MAG assemblies and annotation tables are not linked; deposit to ENA/Zenodo with accession numbers.

Minor language issues – several long sentences and tense shifts (e.g., Introduction lines 55-70); professional polishing will improve readability.

Method details missing – SVM kernel parameters and XGBoost tree depth are reported as “fixed” without concrete values; please list full tuned grid.

Experimental design

Training sample size limited – only 136 reference genomes may not sustain four ML models plus stacking; power analysis or data augmentation suggestion is needed.

Validity of the findings

External validation lacking – all performance metrics are from internal 70/30 split; please add an independent dataset to exclude optimistic bias.

Reviewer 2 ·

Basic reporting

This article deals with an interesting topic about machine learning and the prediction of sporulating firmicutes. The study also presents relevant results according to its hypotheses. The literature references and background are sufficient to assess the contribution of this manuscript.

Experimental design

The research question is well defined, relevant, and meaningful. Also, the authors have performed a rigorous investigation with high technical and ethical standards.
However, the authors failed to provide details about their bioinformatics pipelines. This type of manuscript requires transparency, and the authors should share all pipelines through a reputable repository, such as GitHub.

Validity of the findings

The conclusions of this manuscript are all well stated and linked with the original research question. It is good work, and based on that, it can be improved to be publishable at PeerJ.

Additional comments

Detailed comments and suggestions are in the attached file. Please revise them. I will be glad to revise the improved version of this manuscript.

Annotated reviews are not available for download in order to protect the identity of reviewers who chose to remain anonymous.

---

## Round 0.2 · accepted · Accept

The manuscript can be accepted for publication now.

Reviewer 1 ·

Basic reporting

After reviewing the revised manuscript, I am pleased with the significant improvements made. The authors have effectively addressed the previous concerns, enhancing the overall quality and ensuring it meets publication standards. I fully support its publication and look forward to its contribution to our field.

Experimental design

NA

Validity of the findings

NA

Additional comments

NA

Reviewer 2 ·

Basic reporting

The authors have addressed all comments and suggestions well. Then they need to revise carefully the grammar and syntax of their manuscript.

Experimental design

The authors have addressed all comments and suggestions well. Then they need to revise carefully the grammar and syntax of their manuscript.

Validity of the findings

The authors have addressed all comments and suggestions well. Then they need to revise carefully the grammar and syntax of their manuscript.

Additional comments

The authors have addressed all comments and suggestions well. Then they need to revise carefully the grammar and syntax of their manuscript.